# Nicotinamide adenine dinucleotides are associated with distinct redox control of germination in *Acer* seeds with contrasting physiology

**Shirin Alipour**[1], **Karolina Bilska**[1], **Ewelina Stolarska**[1], **Natalia Wojciechowska**[1,2], **Ewa Marzena Kalemba**[1] *

**1** Institute of Dendrology, Polish Academy of Sciences, Kórnik, Poland, **2** Department of General Botany, Institute of Experimental Biology, Faculty of Biology, Adam Mickiewicz University, Uniwersytetu Poznańskiego 6, Poznań, Poland

* kalemba@man.poznan.pl

**Data Availability Statement:** All relevant data are within the paper and its Supporting Information files

## Abstract

Seed germination is a complex process enabling plant reproduction. Germination was found to be regulated at the proteome, metabolome and hormonal levels as well as via discrete post-translational modification of proteins including phosphorylation and carbonylation. Redox balance is also involved but less studied. *Acer* seeds displaying orthodox and recalcitrant characteristics were investigated to determine the levels of redox couples of nicotinamide adenine dinucleotide (NAD) phosphate (NADP) and integrated with the levels of ascorbate and glutathione. NAD and NADP concentrations were higher in Norway maple seeds and exceptionally high at the germinated stage, being the most contrasting parameter between germinating *Acer* seeds. In contrast, NAD(P)H/NAD(P)$^+$ ratios were higher in sycamore seeds, thus exhibiting higher reducing power. Despite distinct concentrations of ascorbate and glutathione, both seed types attained in embryonic axes and cotyledons had similar ratios of reduced/oxidized forms of ascorbate and half-cell reduction potential of glutathione at the germinated stage. Both species accomplished germination displaying different strategies to modulate redox status. Sycamore produced higher amounts of ascorbate and maintained pyridine nucleotides in reduced forms. Interestingly, lower NAD(P) concentrations limited the regeneration of ascorbate and glutathione but dynamically drove metabolic reactions, particularly in this species, and contributed to faster germination. We suggest that NAD(P) is an important player in regulating redox status during germination in a distinct manner in Norway maple and sycamore seeds.

## Introduction

Germination is one of the critical phases in the plant life cycle, including imbibition of water by quiescent dry seed (phase I), re-initiation of metabolic processes (phase II) and phase III expansion of the embryo and emergence of the radicle [1, 2]. Germination *sensu stricto*

**Funding:** This research was funded by the National Science Center (Poland), grant No. 2015/18/E/NZ9/00729. The Institute of Dendrology of the Polish Academy of Sciences provided support in the form of salary and scholarships for all authors who are affiliated to the Institute. The specific roles of these authors are articulated in the 'author contributions' section. The funders had no role in study design, data collection and analysis, decision to publish, or preparation of the manuscript.

**Competing interests:** The authors have declared that no competing interests exist.

involves first and second phases [1, 2]. The germination process depends on multiple physiological, biochemical and molecular interactions involving signaling and homeostasis at the proteome, transcriptome and metabolome levels (reviewed in Rajjou et al. [3]). Among reactive oxygen species (ROS), hydrogen peroxide ($H_2O_2$) has documented signaling effects [4]. However, a dynamic balance between ROS signaling and oxidative damage is required for the regulation of germination [5]. ROS signals are integrated with hormone signals to control plant developmental processes [6–11]. Whole plant growth and development, including root growth and architecture, which occur during germination, is under redox control [12, 13]. The plant antioxidant system includes many mechanisms of the defense system against oxidative stress [14]. The plant-specific ascorbate-glutathione cycle seems to play a decisive role in redox regulation [15].

Ascorbate (Asc) and glutathione are multifunctional metabolites comprising the cellular redox buffering system, which is required for both ROS scavenging and plant development [15, 16]. Specific functions are assigned to the ascorbate-glutathione cycle in the regulation of metabolism based on cycled switching between the redox couples of Asc, glutathione and nicotinamide adenine dinucleotide (NAD) phosphate (NADP) [17]. The Asc pool consists of ascorbic acid (AsA) and dehydroascorbate (DHA), referring to the reduced and oxidized forms, respectively. The glutathione pool involves reduced glutathione (GSH) and its oxidized form (GSSG). AsA is regenerated by dehydroascorbate reductase (DHAR), which consumes GSH and reduced NAD (NADH). GSH is regenerated by glutathione reductase (GR) using reduced NADP (NADPH) as a cofactor. The ascorbate-glutathione cycle (also called the Foyer–Asada–Halliwell cycle) was found to play an important role in plants in cellular redox environment and in the growth and development of plants [18–20]. Interestingly, ascorbate and glutathione metabolism was reported to be important in seed maturation, drying and germination of desiccation-tolerant (orthodox) and desiccation-sensitive (recalcitrant) seeds [21–23]. Desiccation tolerance acquisition and loss are seed traits especially interesting in terms of redox regulation because the activity of the ascorbate-glutathione cycle differs between orthodox and recalcitrant seeds [21, 23]. More precisely, orthodox seeds display higher activity of all enzymes involved in ascorbate-glutathione cycle. Redox homeostasis is essential for surviving desiccation and further germination [19, 24–26]. Redox changes indicated by a half-cell reduction potential of glutathione were considered important in germinating seeds [27, 28] including seeds with a decreased capacity for ascorbate synthesis [27].

Many of the abovementioned studies focused on changes in ascorbate and glutathione content and the activity of enzymes of the ascorbate-glutathione cycle, whereas the level of NAD (P), the endogenous cofactor that can limit enzymatic reactions, has been neglected. NADH and NADPH are important signaling molecules in plants [29, 30], and together with adenosine triphosphate, NADH and NADPH provide reducing power for the ascorbate-glutathione cycle [31–34]. NAD is a coenzyme associated with catabolic pathways [13, 35–38], whereas NADP is the terminal electron acceptor in biosynthetic processes [39, 40]. The balance between NADH and NADPH and their corresponding oxidized forms $NAD^+$ and $NADP^+$ is necessary for cell survival [41] by controlling redox reactions [42, 43]. $NADH/NAD^+$ and $NADPH/NADP^+$ play a central role in regulating plant growth, germination initiation and time to germination onset [44–46]. Shifting from the NADH to $NAD^+$ redox state governs Arabidopsis pollen germination time [46, 47]. However, the link between nicotinamide nucleotides and germination in plant seeds remains fairly unknown. Concentrations of NAD(P) redox couples might be converted to many informative physiological indices. The anabolic redox charge (ARC) and catabolic redox charge (CRC) characterize NAD(P)-driven metabolism [48], reflecting decomposition and synthesis reactions in germinating seeds [49]. NAD kinase (NADK) converts NAD to NADP, and its activity is expressed as phosphorylation capacity based on

concentrations of redox forms of pyridine nucleotides [50]. In Arabidopsis, NADK1 is situated in the cytosol, NADK2 in chloroplasts and NADK3 in peroxisomes [51]. Interestingly, NADK1 uses NAD$^+$, whereas NADK3 uses NADH for phosphorylation. Concentrations of NAD seem to be related to seed dormancy [42]. Thus, the NAD/NADP ratio correlates with depth of seed dormancy in Arabidopsis [52]. Additionally, the reduction power driven by NAD(P) might be calculated [32] and used to contrast the state of the energy balance in orthodox and recalcitrant seeds [53].

Norway maple (*Acer platanoides* L.) and sycamore (*Acer pseudoplatanus* L.) seeds are physiologically contrasted and used as a model system to investigate seed development [21, 54], seed dormancy [55–57], seed drying and desiccation [26, 53, 58–61]. To date, studies on the involvement of pyridine nucleotides in the antioxidant machinery during germination in *Acer* seeds are lacking. Desiccation tolerance acquisition benefits Norway maple seeds during seed maturation and desiccation. Recently, NAD(P) concentrations and their redox status were reported to be involved in desiccation tolerance in the seeds of the two *Acer* seeds because Norway maple contained high NADPH concentrations, accumulated NAD$^+$, and displayed a low and constant NAD(P)H/NAD(P)$^+$ ratio in contrast to sycamore seeds [53]. Interestingly, during the germination process, desiccation tolerance is progressively lost [62, 63]. Thus, it was necessary to determine whether Norway maple seeds are still better protected at germination stages in terms of redox regulation emphasized at the concentrations of NAD(P), Asc, and glutathione redox couples and their redox states. Our studies provided novel information about redox regulation of the germination process in *Acer* seeds. More precisely, NAD(P) has a superior role in limiting ascorbate and glutathione regeneration. Our studies revealed that Norway maple and sycamore seeds strongly differed in NAD(P), Asc, glutathione concentrations, NAD(P)-dependent metabolic activity and reducing power during all germination stages, highlighting different redox control of the germination process in the two *Acer* species, presumably by providing distinct strategies that are discussed in this report.

## Material and methods

### Seed material

Seeds of Norway maple (*Acer platanoides*) and sycamore (*Acer pseudoplatanus*) were collected from the individual trees growing in Kórnik (Western Poland) at the 23$^{rd}$ and 24$^{th}$ weeks after flowering. Analyses were performed for several stages of germination starting with dried seeds. Seeds were dried (D) to 10% water content (WC) for Norway maple and 30% WC for sycamore. Seeds were hydrated for 24 h, and imbibed seeds (I) were placed on wet paper towels in plastic boxes and kept at 3˚C (cold stratification). Every week, the wet paper towels were replaced with new ones, and the decayed seeds were removed. Every two weeks (sycamore: 2–8 w) and three weeks (Norway maple: 3–9 w), seeds representing the successive stages of germination were taken for analyses. Seeds were assayed as germinated (G) when the radicle protruded to 5 mm above the seed testa. At the 9$^{th}$ (sycamore) and 12$^{th}$ (Norway maple) weeks after imbibition, embryonic axes were protruding outside the seed coat and constituted the last stage–germinated (G) seeds [64]. Seeds were highly viable because they germinated in nearly 100% of both species.

### Preparation of extract

To determine the ascorbate pool, the glutathione pool and NAD(P) content extracts were prepared according to the method described by Queval and Noctor [65]. Embryonic axes and cotyledons were ground in 0.2 M HCl. Twenty embryonic axes and five cotyledons were taken for one sample. The homogenates were centrifuged for 10 min at 4˚C and 14 000 rpm. The

supernatant intended for NAD(P) determination was incubated for 2 min at 100˚C, and after cooling, the pH of the samples was adjusted to 6–7. The extract intended for Asc and glutathione determination was adjusted to pH 4.5–5. The reaction results were measured using an Infinite M200 PRO (TECAN) plate reader and Magellan software.

1. **Determination of NAD(P)H.** A reaction mixture was prepared with the following composition: 100 μl of 10 mM HEPES/2 mM ethylenediaminetetraacetic acid (EDTA) (pH 7.5), 20 μl of 1.2 mM 2,6-dichlorophenolindophenol, 10 μl of 10 mM phenazine methosulfate, 30 μl of $H_2O$ and 20 μl of neutralized extracts. Two units (U) of glucose-6-phosphate dehydrogenase and 10 μl of 10 mM glucose-6-phosphate were added to measure NADP, whereas for the measurement of NAD, 25 U alcohol dehydrogenase and 15 μl of ethanol were added. The kinetic measurements were performed at 600 nm. The levels of reduced and oxidized forms of NAD(P) were calculated from calibration curves prepared using NADPH, $NADP^+$, NADH and $NAD^+$ (Sigma-Aldrich, St. Louis, MO, USA) as standards.

2. **Determination of ascorbate.** For ascorbate determination, the method that enables the measurement of even an extremely low amount of AsA was chosen [26, 65, 66]. Total ascorbate (Asc; AsA + DHA) was measured by reducing the extract with 25 mM dithiothreitol at pH 4.7. AsA was measured in neutralized extracts by its absorption light at 265 nm in a slightly acidic environment. The measurements of absorbance were performed in 0.1 mM acetic acetate buffer containing 5 mM EDTA. The determination of DHA was calculated by subtracting free AsA from the total Asc.

3. **Determination of glutathione.** The neutralized extract was treated with 2-vinylpyridine (2-VP) for 30 min at room temperature (RT) and centrifuged twice for 15 min at 4˚C and 14,000 rpm. The reaction mixture contained 120 mM $NaH_2PO_4$/10 mM EDTA pH 7.5, 12 mM 5,5'-dithiobis(2-nitrobenzoic acid), 10 mM NADPH, MQ water and extract (to measure total glutathione, GSH + GSSG) or 2-VP-treated extract (to determine only oxidized form GSSG), and glutathione reductase (0.2 U). The measurements were performed at 412 nm. Calculations were based on calibration curves prepared using GSSG and GSH (Sigma-Aldrich) as standards.

## NAD(P)-originated physiological indices

ARC was calculated using the equation ARC = NADPH/(NADPH+$NADP^+$), whereas CRC was calculated using the equation CRC = NADH/(NADH+$NAD^+$) [48]. Phosphorylation capacity of NADK1 and NADK3 was calculated as product to substrate ratio $NADP^+$/$NAD^+$ and NADPH/NADH, respectively. The depth of dormancy was expressed as the NAD/NADP ratio [52]. The equation (NADH + NADPH)/($NAD^+$ + NADH) + ($NADP^+$ + NADPH) described by Quebedeaux [32] was used to calculate the reduction power driven by NAD(P) and was expressed as a ratio in the range 0–1.

## Redox potential

The redox potential (*E*) was calculated using the Nernst equation: $E = E_0 - (RT/nF)\log([red]/[ox])$ where $E_0$, standard half-cell reduction potential at pH 7 ($E_{0\ GSSG/2GSH}$ = −240 mV, $E_{0\ DHA/AsA}$ = 80 mV); R, gas constant (8.314 $JK^{-1} mol^{-1}$); *T*, temperature [K]; n, number of electrons involved in the reaction; F, Faraday constant (9.6485 $10^4 C\ mol^{-1}$); red, molar concentration of reduced form; ox, molar concentration of oxidized form [67]. $E_0$ was adjusted to $E_{pH}$ as described in Schafer and Buettner [68].

## Statistical analyses

All analyses were performed for three biological replicates. Lack of statistically significant differences was indicated with the same letters [one-way analysis of variance (ANOVA), followed by Tukey's test at P = 0.05]. The relationships between individual parameters were assessed using the Pearson correlation coefficient analysis. Proportional data were transformed prior to analysis using the arcsine transformation. R statistical software was used to calculate Pearson's correlation coefficients using R statistical software [69]. The corrplot package was utilized to create the correlation matrices [70].

# Results

## Levels of NAD

NAD levels were clearly higher in Norway maple seeds than in sycamore seeds, particularly in embryonic axes. Imbibition for 24 h halved the NADH and NAD$^+$ levels in Norway maple embryonic axes, whereas similar levels of NADH and NAD$^+$ were detected in dry and in imbibed cotyledons (Fig 1). Similar levels of NADH and NAD$^+$ levels were reported in Norway maple embryonic axes from the imbibition stage up to the 9$^{th}$ week after imbibition (WAI), then increased and reached a maximum level at the germinated stage. In cotyledons, a declining trend in NAD concentrations was reported. Interestingly, at the germinated stage, NADH and NAD$^+$ contents were fourfold and sixfold

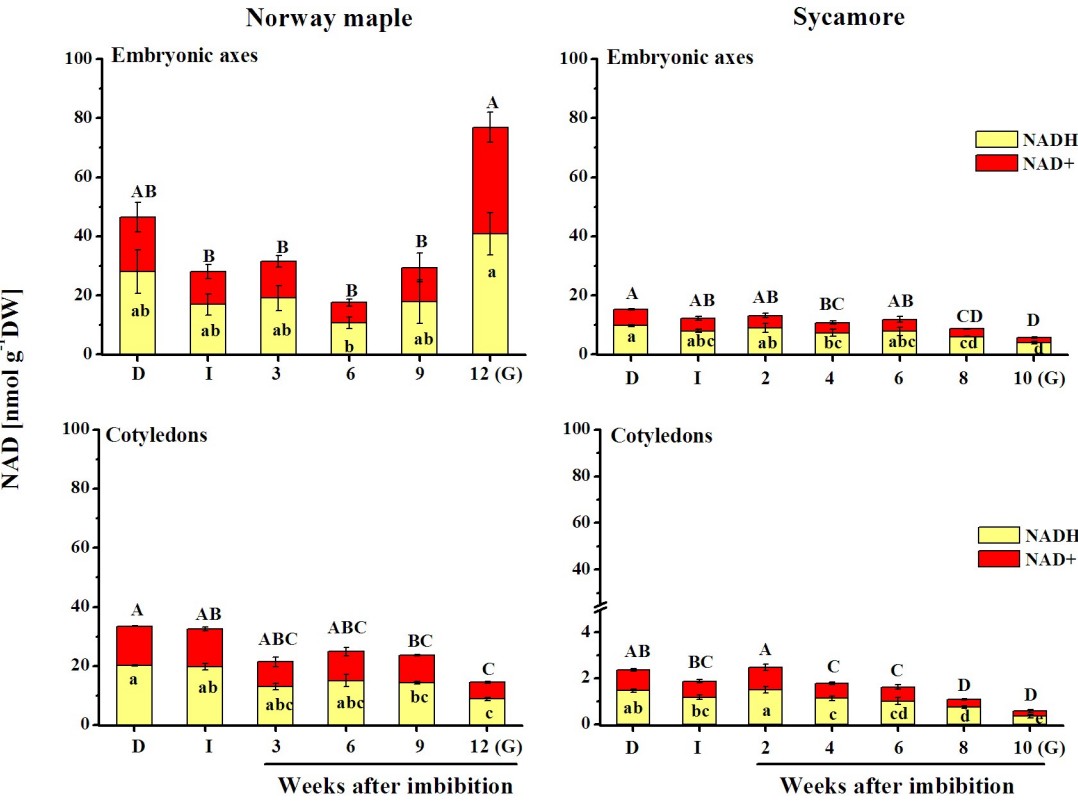

**Fig 1. Levels of reduced (NADH) and oxidized (NAD$^+$) forms of nicotinamide adenine dinucleotide (NAD) detected in dry and germinating Norway maple and sycamore seeds.** (D) Dry seeds; (I) Imbibed seeds; (G) Germinated seeds. Data represent the mean ± standard deviation of three independent replicates. Statistically significant differences are indicated with different letters (one-way ANOVA followed by Tukey's test at p ≤ 0.05). The capital letters refer to the oxidized form.

higher, respectively, in Norway maple embryonic axes compared to cotyledons. Almost linear decreases in NADH and NAD$^+$ were observed in sycamore seeds. The levels of NADH and NAD$^+$ were up to 6 times higher in embryonic axes than in cotyledons. Interestingly, NADH levels were approximately 12 times higher in embryonic axes than in cotyledons at the germinated stage. In summary, germinating Norway maple seeds exhibited higher concentrations of NAD, particularly in embryonic axes at the germinated stage.

## Levels of NADP

Norway maple contained higher NADP concentrations than sycamore, particularly in cotyledons (Fig 2). In the embryonic axes, NADPH levels were similar up to the 6$^{th}$ WAI; however, NADP$^+$ levels were completely different between Norway maple and sycamore. Norway maple embryonic axes contained triple NADP levels at the germinated stage. Interestingly, up to ten times higher NADP levels were detected in sycamore embryonic axes than in cotyledons. NADP gradually decreased in cotyledons of both *Acer* species. Extremely low levels of NADPH and NADP$^+$ were reported at the germinated stage in sycamore seeds. At this stage, NADPH content was 18-fold higher in embryonic axes compared to cotyledons. Sycamore seeds displayed lower NADP levels, which decreased upon germination, whereas embryonic axes of Norway maple embryonic axes at the germinated stage showed a significant increase in NADP concentration.

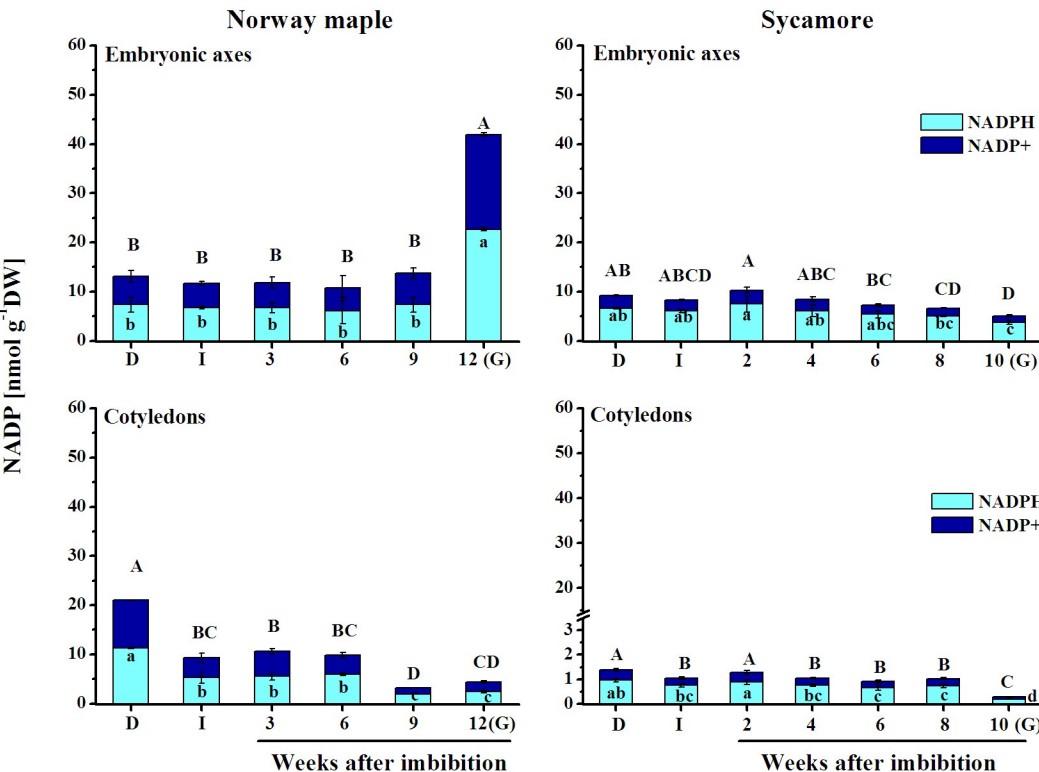

**Fig 2. Levels of reduced (NADPH) and oxidized (NADP$^+$) forms of nicotinamide adenine dinucleotide phosphate (NADP) detected in dry and germinating Norway maple and sycamore seeds.** (D) Dry seeds; (I) Imbibed seeds; (G) Germinated seeds. Data represent the mean ± standard deviation of three independent replicates. Statistically significant differences are indicated with different letters (one-way ANOVA followed by Tukey's test at p ≤ 0.05). The capital letters refer to the oxidized form.

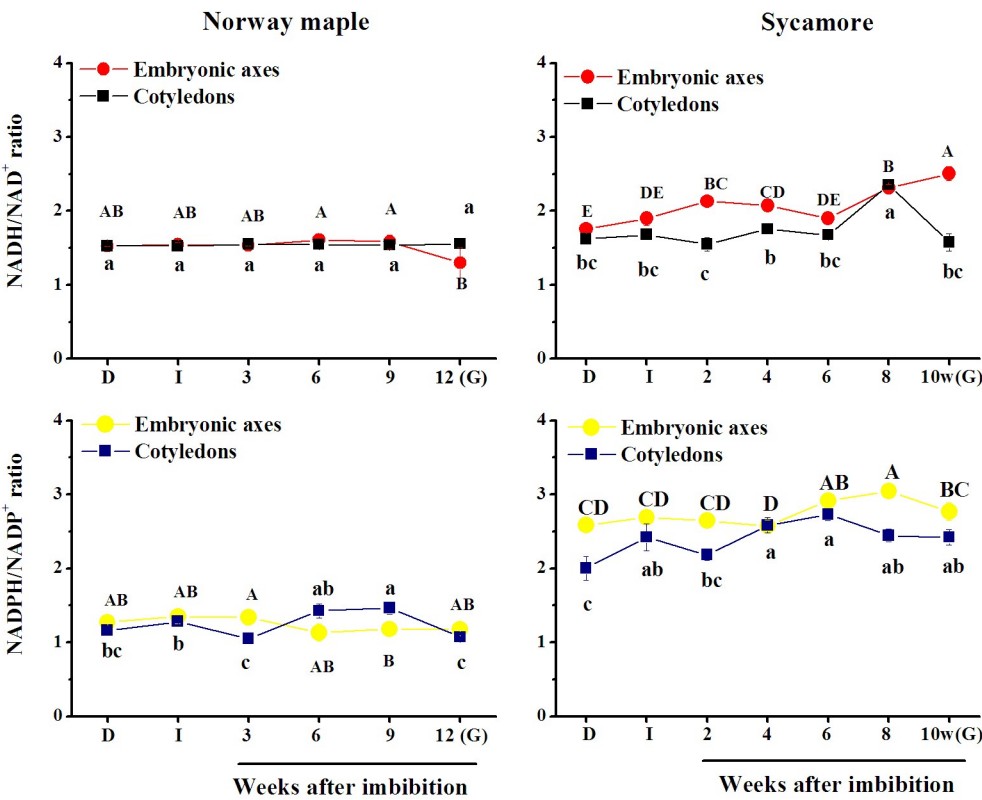

**Fig 3. Changes in the ratios of reduced (NAD(P)H) to oxidized (NAD(P)⁺) forms of nicotinamide adenine dinucleotide (NAD) and its phosphate (NADP) detected in dry and germinating Norway maple and sycamore seeds.** (D) Dry seeds; (I) Imbibed seeds; (G) Germinated seeds. Data represent the mean ± standard deviation of three independent replicates. Statistically significant differences are indicated with different letters (one-way ANOVA followed by Tukey's test at p ≤ 0.05). The capital letters refer to the embryonic axes.

## NAD(P)H/NAD(P)⁺ ratios

The NADH/NAD⁺ ratio was lower in Norway maple seeds than in sycamore (Fig 3). The NADH/NAD⁺ ratio slightly decreased in the embryonic axes of Norway maple at the germinated stage. In sycamore embryonic axes, an increasing trend of the NADH/NAD⁺ ratio peaked at the germinated stage, whereas in cotyledons, the peak of NADH/NAD⁺ ratio was reported at the 8th WAI and then rapidly declined at the germinated stage. The NADH/NAD⁺ ratio was clearly higher in sycamore seeds than in Norway maple throughout the germination process (Fig 3).

The NADPH/NADP⁺ ratio remained relatively unchanged up to the germinated stage of Norway maple embryonic axes, whereas slightly differed in cotyledons, reaching extremely low levels at the 3rd WAI and at the germinated stage. A similar pattern of changes was reported in sycamore seeds, being at least twofold higher than in Norway maple. The NADPH/NADP⁺ ratio was the highest in sycamore embryonic axes at the 8th WAI, reaching 3.

## NAD(P)-originated physiological indices

Germination-driven metabolism might be characterized via NAD(P)-derived parameters, including ARC and CRC. Both ARC and CRC were constant throughout germination in Norway maple seeds (Fig 4A). ARC was considerably higher in sycamore seeds. Additionally, in sycamore, an increasing tendency of CRC was reported. CRC reached the highest values at the final stages of germination and peaked in elongated embryonic axes. CRC

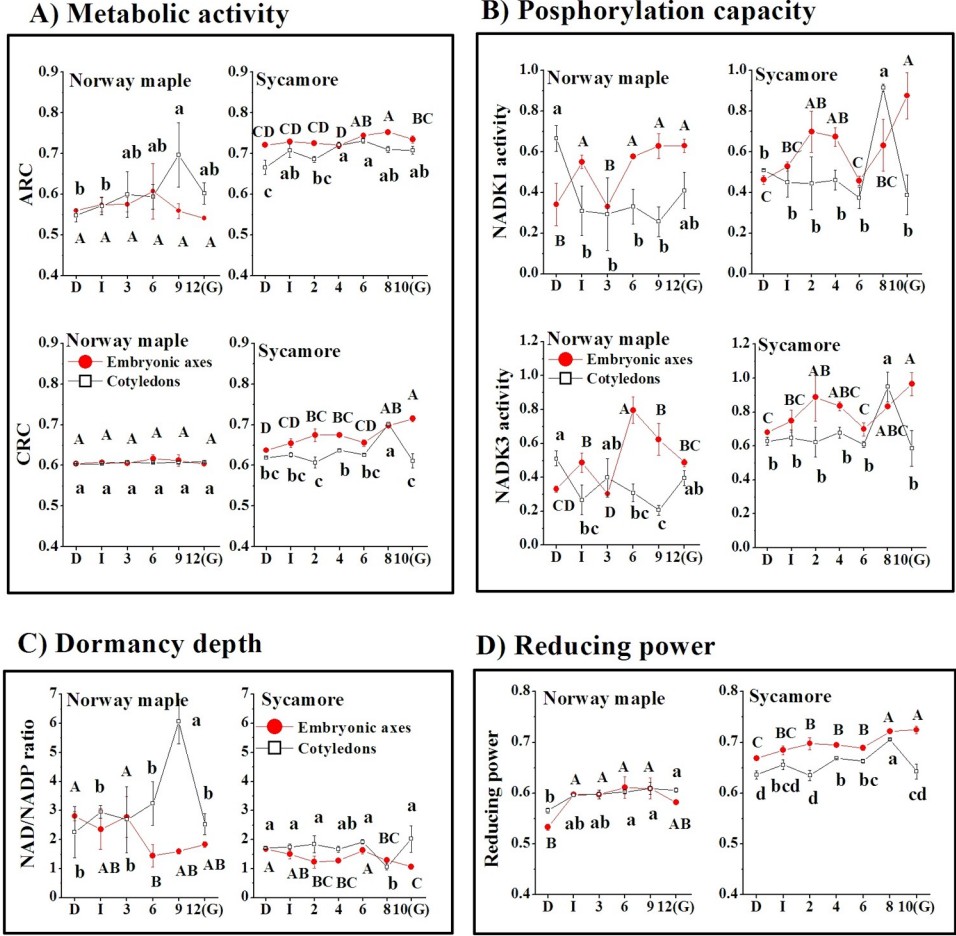

**Fig 4. NAD(P)-originated physiological indices.** A) Metabolism-related catabolic redox charge (CRC) and anabolic redox charge (ARC); B) phosphorylation capacity of NADK1 and NADK3; C) dormancy depth; D) reducing power. (D) Dry seeds; (I) Imbibed seeds; (G) Germinated seeds. Data represent the mean ± standard deviation of three independent replicates. Statistically significant differences are indicated with different letters (one-way ANOVA followed by Tukey's test at p ≤ 0.05). The capital letters refer to embryonic axes.

peaked in cotyledons at the 8th WAI just before the accomplishment of germination. Phosphorylation capacity is the result of NADK1 activity converting NAD+ to NADP+ or NADK3 activity converting NADH to NADPH (Fig 4B). Phosphorylation capacity peaked at 0.7 in Norway maple embryonic axes at the 6th WAI. NADK1 activity in sycamore embryonic axes displayed identical values earlier, between the 2nd and 4th WAI, and further increased to 0.9 in the germinated stage. Sycamore cotyledons exhibited a peak of NADK1 at the 8th WAI, being 2-fold higher than at other germination stages. The phosphorylation capacity of NADK3 displayed similar patterns in seeds of both *Acer* species; however, its activity was clearly higher in sycamore seeds. The ratio of NAD/NADP pools can be used as an indicator of the depth of dormancy. Norway maple seeds exhibited higher NAD/NADP ratios related to deeper dormancy than sycamore seeds (Fig 4C). A clearly decreasing ratio in embryonic axes is a sign of the gradual alleviation of dormancy in both species. The nicotinamide redox charge highlights the nicotinamide-based reducing power, which was reported to be higher in sycamore seeds (Fig 4D). Except for the dry stage, embryonic axes and cotyledons of Norway maple exhibited quite similar values, whereas sycamore seeds

displayed distinct values that were higher in embryonic axes. The NAD(P)-based physiological indices related to metabolism and dormancy depth showed clear differences between the *Acer* species that were studied. Interestingly, sycamore seeds displayed higher CRC, higher reducing power and higher activity of putative NADK3 at all germination stages.

## Levels of ascorbate

The total ascorbate pool was markedly higher in sycamore seeds than in Norway maple seeds, particularly in the embryonic axes, in which nearly sevenfold higher levels were reported, whereas in cotyledons, except the germinated stage, doubled concentrations of Asc were detected (Fig 5). In Norway maple seeds, the Asc level was elevated at the imbibed and germinated stages. In contrast, in sycamore seeds, the Asc level substantially decreased at the latter stage. The highest Asc level in sycamore cotyledons was observed at the dry seed stage, and a decreasing trend was detected up to the germinated stage.

Norway maple seeds displayed a higher AsA/DHA ratio that changed dynamically during germination stages, except imbibition, compared to sycamore seeds (Fig 5). However, at the germinated stage, the AsA/DHA ratio was relatively similar between embryonic axes and cotyledons in both species. Dry and imbibed seeds of Norway maple differed substantially in the AsA/DHA ratio between embryonic axes and cotyledons; however, at the germinated stage,

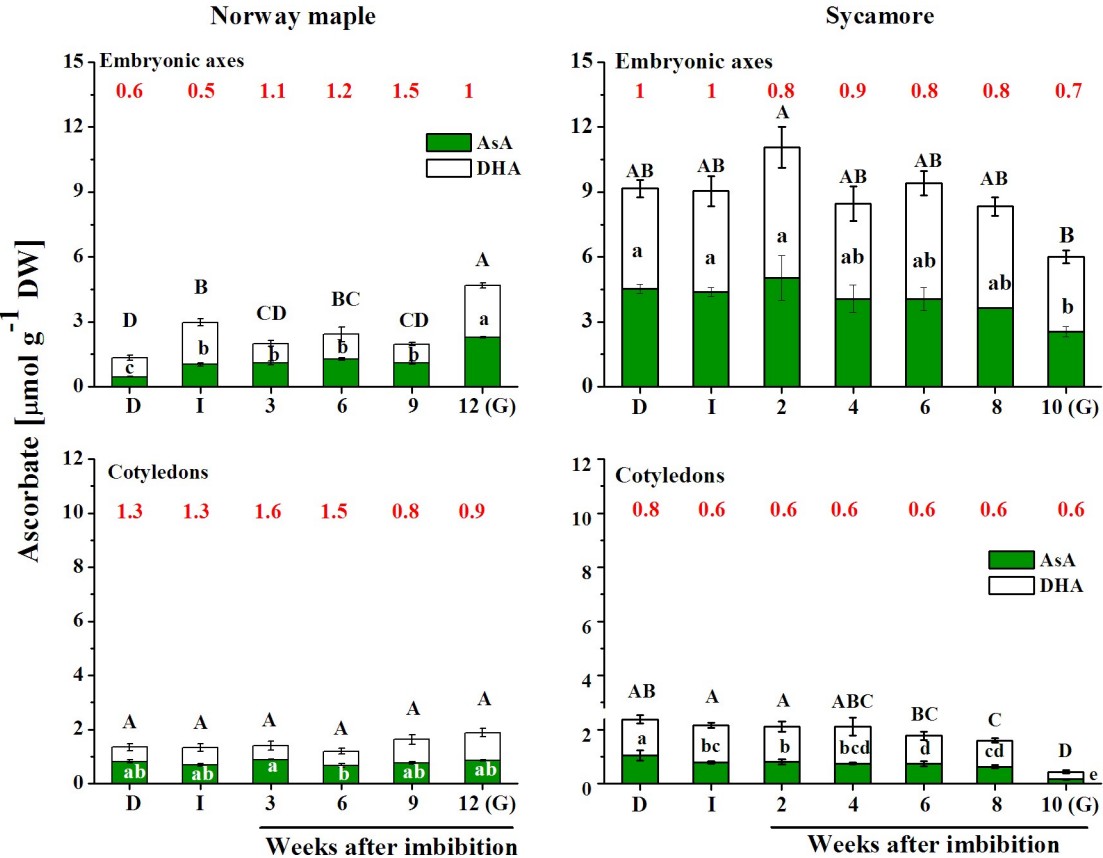

**Fig 5. Changes in the levels of the reduced (AsA) and oxidized (DHA) forms of ascorbate and their ratio (red font) reported in the embryonic axes and cotyledons of dry and germinating Norway maple and sycamore seeds.** (D) Dry seeds; (I) Imbibed seeds; (G) Germinated seeds. Data are the means of three independent replicates ± std. Statistically significant differences are indicated with different letters (one-way ANOVA, followed by Tukey's test at p < 0.05). The capital letters refer to DHA.

this ratio was more similar in both seed tissues. AsA/DHA ratios lower than 1 indicate that DHA dominates in the Asc pool, which was reported in sycamore seeds.

## Levels of glutathione

The glutathione pool was comparable in the embryonic axes of both species at the initial stages of germination but differed at the germinated stage, being over 4-fold higher in the Norway maple than in the sycamore (Fig 6). Sycamore cotyledons contained less glutathione than Norway maple cotyledons and substantially decreased at the germinated stage.

## Redox potentials

The most distinct $E_{DHA/AsA}$ between Norway maple embryonic axes and cotyledons was reported at dry and imbibed stages, after which, $E_{DHA/AsA}$ was comparable. In the sycamore, $E_{DHA/AsA}$ was higher in the cotyledons and similar in dry and germinating seeds. In both species, $E_{GSSG/2GSH}$ was higher in the cotyledons than in the embryonic axes, except the germinated stage, at which $E_{GSSG/2GSH}$ reached identical values in sycamore (Fig 7). The pattern of changes in the $E_{GSSG/2GSH}$ was dynamic and similar in the embryonic axes and cotyledons of Norway maple seeds. Interestingly, $E_{GSSG/2GSH}$ was maintained relatively stable in sycamore embryonic axes reflecting that approximately 80% of the glutathione was reduced.

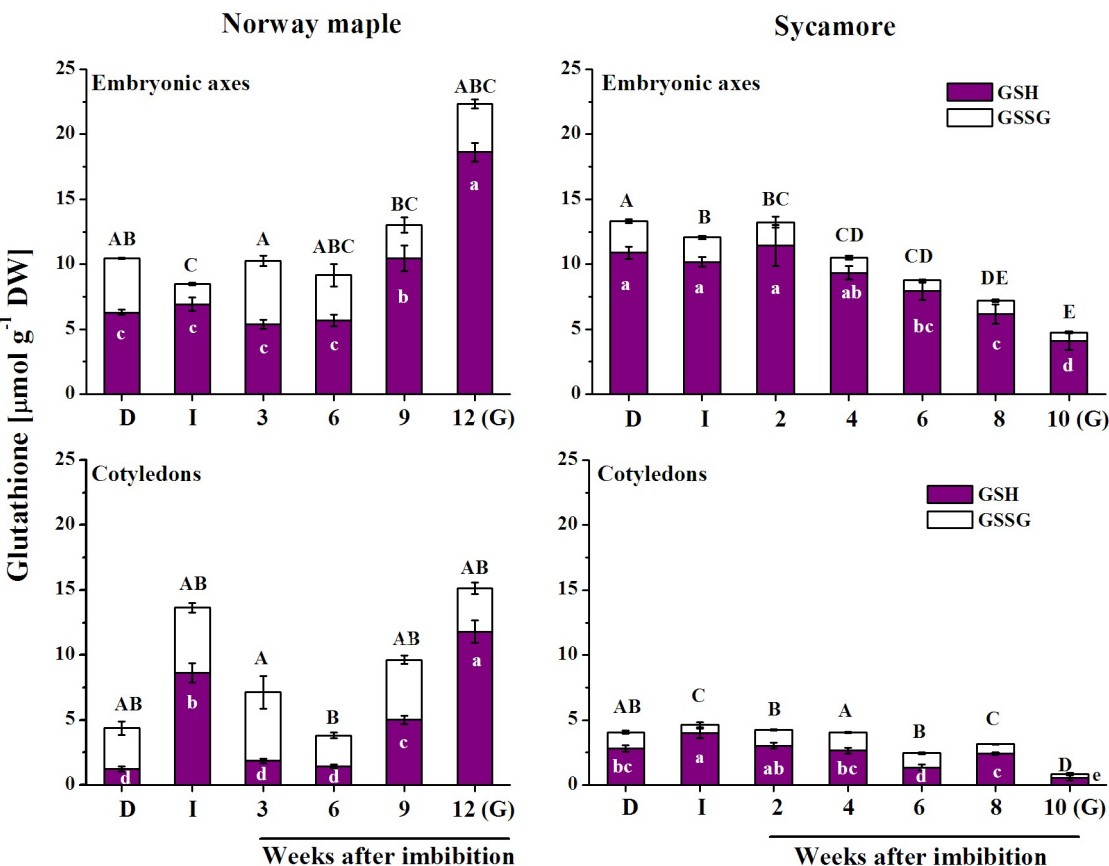

**Fig 6. Changes in the levels of the reduced (GSH) and oxidized (GSSG) forms of glutathione reported in the embryonic axes and cotyledons of dry and germinating Norway maple and sycamore seeds.** (D) Dry seeds; (I) Imbibed seeds; (G) Germinated seeds. Data are the means of three independent replicates ± std. Statistically significant differences are indicated with different letters (one-way ANOVA, followed by Tukey's test at p < 0.05). The capital letters refer to GSSG.

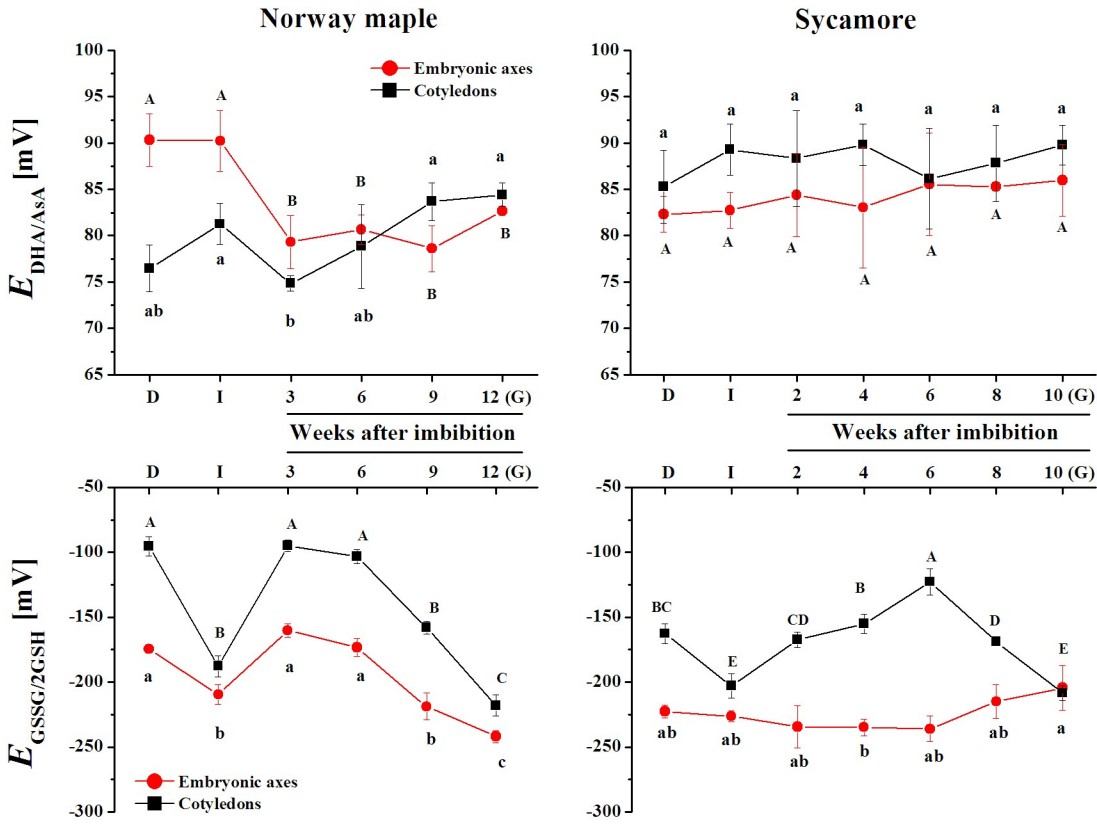

**Fig 7. Changes in the levels of the half-cell reduction potential of glutathione ($E_{GSSG/2GSH}$) and ascorbate ($E_{DHA/AsA}$) reported in the embryonic axes and cotyledons of dry and germinating Norway maple and sycamore seeds.** (D) Dry seeds; (I) Imbibed seeds; (G) Germinated seeds. Data are the means of three independent replicates ± std. Statistically significant differences are indicated with different letters (one-way ANOVA, followed by Tukey's test at p < 0.05).

On the basis of the results obtained in this study we propose a scheme showing the most contrasting characteristics reported in orthodox and recalcitrant seeds in respect to redox regulation during *Acer* seed germination (Fig 8).

## Correlations

The concentrations of NAD(P) redox couples were positively correlated with the levels of AsA, DHA, Asc, GSH and GSSG in sycamore seeds and Norway maple embryonic axes (S1A Fig). Concentrations of NAD(P) were also linked in sycamore embryonic axes, with the AsA/DHA ratio being positively correlated with concentrations of NADPH in Norway maple cotyledons (S1B Fig). The NADPH/NADP⁺ ratio was negatively correlated with the levels of DHA in Norway maple embryonic axes and with AsA and Asc in sycamore seeds. The NADH/NAD⁺ ratio was negatively correlated with the levels of DHA, Asc and GSH in Norway maple embryonic axes, whereas in sycamore embryonic axes, the ratio was negatively correlated with the levels of GSH, GSSG and the AsA/DHA ratio (S1C Fig). CRC and ARC were negatively correlated with the glutathione redox couple and the AsA/DHA ratio in sycamore embryonic axes. Additionally, the levels of AsA were negatively correlated with CRC levels in sycamore embryonic axes and ARCs in cotyledons, and the latter were also correlated with the AsA/DHA ratio (S1D Fig). The phosphorylation capacity of NADK1 was correlated with GSH levels in the embryonic axes of both species but in an opposite manner. Additionally, in Norway maple

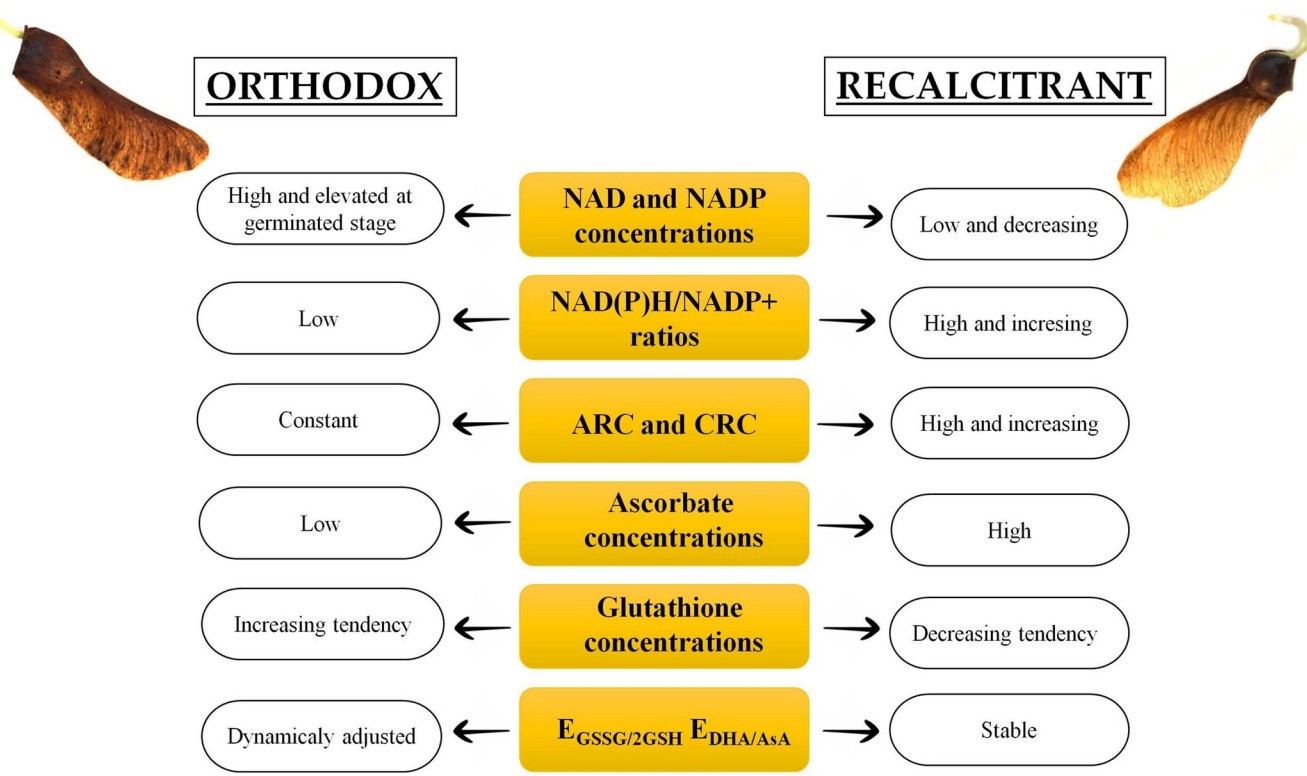

**Fig 8. A scheme of strategies describing distinct redox regulation during germination of orthodox (Norway maple) and recalcitrant (sycamore) seeds.**

embryonic axes, the phosphorylation capacity of NADK1 was negatively correlated with GSSG level and positively correlated with AsA. The dormancy depth expressed as the ratio of NAD/NADP pools displayed a negative correlation with the AsA/DHA ratio in Norway maple embryonic axes and a positive correlation with GSH levels in sycamore embryonic axes. Importantly, $E_{GSSG/2GSH}$ was negatively correlated with concentrations of both NAD and NADP redox couples uniquely in sycamore seeds (S1C and S1D Fig).

## Discussion

Our recent studies revealed that regulation of methionine redox homeostasis differs in germinating Norway maple and sycamore seeds [64], which prompted us to investigate the redox balance in *Acer* seeds in more detail. Redox homeostasis during cell growth is the result of the interplay between NAD(P) pools and ascorbate and glutathione [71]. More precisely, the interplay implies that ascorbate and glutathione are the cutting edge of the NAD(P)-based buffering system [72]. Thus, it was necessary to investigate whether concentrations of NAD(P) redox couples might affect the regeneration of AsA and GSH during germination of orthodox and recalcitrant *Acer* seeds.

### Regulation of pyridine nucleotides in germinating *Acer* seeds

NAD(P) influences all metabolic pathways as a versatile electron source [73], and together with ATP, NAD(P) is the main energy source for plant cells [74]. Throughout germination, Norway maple seeds contained higher NAD and NADP pools than sycamore seeds (Figs 1 and 2). Increased levels of NAD redox couples were reported in embryonic axes of orthodox-

seeded *Avena sativa* during the germination phase [75]. Recalcitrant sycamore seeds displayed the opposite pattern of changes. Importantly, concentrations of NAD and NADP were the most contrasted between *Acer* species at the germinated stage, being 8-13-fold higher in Norway maple embryonic axes than in sycamore embryonic axes and 15-26-fold higher in Norway maple cotyledons than in sycamore cotyledons (Figs 1 and 2), indicating that regulation of redox status is vastly different in the two *Acer* species. Norway maple and sycamore produce seeds differing in response to water loss [58]. A linear increase in water content (WC) was observed in germinating sycamore seeds, whereas in Norway maple seeds changes in WC fitted the classical triphasic seed germination model [64]. Distinct absorption of water might be also related to differential behavior of *Acer* seeds during germination. Comparing NAD(P) concentrations (Figs 1 and 2) and changes in WC in *Acer* seeds [64], which were also used in this study, a strong negative correlation (R = - 0.8) can be noticed particularly for sycamore seeds.

## Involvement of NAD(P) in dormancy regulation

Seed germination is a complex process [1, 2] lasting up to 16 and 20 weeks in sycamore and Norway maple, respectively [76]. In our previous results [64] and in this report, Norway maple started to elongate radicles at the 12[th] WAI, whereas in sycamore seeds, this phenomenon appeared three weeks earlier, in line with other germination studies based on non stored *Acer* seeds [57]. NAD[+] accumulation is an attribute of desiccation tolerance observed in dry orthodox Arabidopsis pollen [46, 77] and desiccated orthodox Norway maple seeds [53]. Importantly, NAD[+] prevents useless metabolic cycling in the dormant state [46]. Accumulated NAD[+] repressed the time of germination and growth of pollen tubes [77]. Thus, a decrease in NAD[+] appeared to be important in the metabolic state transition [46, 77]. A clear decline in NAD[+] was reported only in sycamore seeds (Fig 1) as a signal for germination [46] and was probably associated with faster germination in this species. Higher NAD levels were reported in more dormant Arabidopsis ecotypes [78]; thus, higher NAD concentrations measured in Norway maple seeds (Fig 1) might reflect deeper dormancy in this species compared to sycamore. The breaking of dormancy is emphasized by a depletion of NAD concentration [78]. This phenomenon was easily observed in sycamore seeds (Fig 1). The NAD/NADP ratio may be involved in the determination or control of the depth of seed dormancy [52]. A clearly decreasing ratio in embryonic axes might be a sign of the gradual alleviation of dormancy in both species. In this context, the levels of both pyridine nucleotides and their redox status clearly reflect seed dormancy depth and its alleviation in *Acer* seeds.

## Association of NAD(P) in the regulation of metabolism in germinating *Acer* seeds

NAD redox couples can effectively control metabolic activity during the germination process [77]. Pyridine nucleotides provide a picture of germination-driven metabolism via ARC and CRC [48]. The increase in CRC can initiate germination [49]. CRC increased only in sycamore seeds (Fig 4A). ARC is 0.5 and higher, and higher values manifest readiness to drive biosynthetic reactions [48]. Interestingly, increased ARC in sycamore seeds (Fig 4A) indicated an earlier inception of synthesis reactions in the G2 phase in this species, probably by the elements of the ascorbate-glutathione cycle, as confirmed by the correlations (S1 Fig).

NAD affects NADP levels [38]; thus, NAD homeostasis can affect NADP-dependent reactions [73, 79, 80]. The correlation between pools of NAD and NADP was strong and positive, except in Norway maple cotyledons, indicating that NADKs were active in *Acer* seeds. The phosphorylation capacity of putative NADK1 and NADK3 exhibited similar patterns and

peaked at higher values in sycamore seeds, particularly at final germination stages (Fig 4B). NADKs provide NADP, a substrate for glycolysis and the pentose phosphate pathway (PPP). Considering the NAD(P) content, which was higher in Norway maple and lower in sycamore seeds (Figs 1 and 2), both glycolysis and PPP seem to proceed differentially in *Acer* seeds during germination. Additionally, the activity of NADK3 was higher in sycamore seeds; thus, deeper characterization of metabolic processes related to pyridine nucleotides in germinating *Acer* seeds is needed because reduction and oxidation cycling of NAD(P) are not inadvertent but depend strictly on enzymatic reactions [38].

A low $NADPH/NADP^+$ ratio activates the PPP at the oxidative stage (OPPP) [81]. Thus, the considerably lower but relatively stable ratio reported in Norway maple seeds (Fig 3) indicates that in this species, the OPPP pathway proceeds throughout germination *sensu stricto*. A peak of NADPH in Norway maple embryonic axes (Fig 2) coincided with radicle elongation and is a sign of elevated OPPP activity [49]. In this context, at the $12^{th}$ WAI, germination was advanced in Norway maple seeds and coincided with the increase in NADH content (Fig 1), which was assumed to be a germination accelerator [82]. Highly accumulated pyridine nucleotides in their oxidized forms inhibit seed germination [52]. Thus, higher concentrations of both $NAD^+$ and $NADP^+$ reflected in lower $NAD(P)H/NAD(P)^+$ ratios in Norway maple seeds (Figs 1–3) were associated with the fact that radicle protrusion was initiated several weeks later in Norway maple than in sycamore seeds. Moreover, the $NADH/NAD^+$ ratio is high in dynamically growing tissues [38]. Thus, the higher $NADH/NAD^+$ ratio observed in sycamore seeds (Fig 3) at the time when radicles were visibly elongated is in line with the findings that NADH accelerates germination and that its oxidized form delays this process [52, 82].

## Reducing power in germinating seeds

The nicotinamide redox charge emphasizes the state of the energy balance between the oxidized and reduced nicotinamide nucleotide forms [32]. It is an expression of the reducing power ranging from 0 to 1 being higher in growing tissues and lower than 0.5 in senescing or dormant tissues [31, 32]. Higher levels of metabolically available energy stored in the nicotinamide nucleotide system in germinating sycamore seeds (Fig 4D) might be the effect of domination of the reduced state in nicotinamides as a strategy to compensate for significantly lower concentrations of NAD(P) (Fig 3). Thus, putatively faster redox cycling of pyridine nucleotides in germinating sycamore seeds needs further investigation to confirm it. In general, redox potentials in seed tissues with low redox buffering capacity are more oxidized [27]. More oxidized $E_{GSSG/2GSH}$ and $E_{DHA/AsA}$ were reported in cotyledons of both *Acer* species (Fig 7) indicating that embryonic axes are better protected, whereas cotyledons, which are storage organs, display distinct redox regulatory patterns during germination. The reduction potential for the $NADP^+/NADPH$ half-cell is independent of their concentrations and depends on the ratios of the reduced and oxidized forms[68], which were presented in Fig 3. Ratios calculated for pyridine nucleotides definitely indicate that NAD and NADP are more reduced in sycamore seeds. Additionally, the nicotinamide redox charge, which highlights the nicotinamide-based reducing power, was reported to be higher in sycamore seeds (Fig 4D).

## NADPH contributes to thiol redox status

Thiol redox status contributes to successful germination [83]. Mitochondrial energy metabolism starts rapidly at imbibition of orthodox seeds [83]. NADPH provides reducing energy in ROS scavenging [84] and enables GSH regeneration. Higher NADPH levels detected in germinated Norway maple seeds (Fig 2) indicated that ROS homeostasis and GSH regeneration were more efficient in orthodox seeds particularly at this stage. NADPH content was similar in

imbibed embryonic axes of both *Acer* species (Fig 2). Thus, GR may be more active at the imbibed stage because the embryonic axes of desiccated Norway maple seeds exhibit two-fold higher GR activity than dry sycamore seeds [21]. The highly positive correlation between NAD(P)H concentrations and AsA and GSH concentrations (S1 Fig) suggests that NADH and NADPH might be the limiting factors in sycamore seeds for the efficiency of the ascorbate-glutathione cycle, in which the regeneration of AsA and GSH depends on the availability of a reduced form of pyridine nucleotides [20].

## Regulation of ascorbate and glutathione in germinating *Acer* seeds

DHAR catalyzes GSH-dependent reduction of DHA only in the presence of reduced pyridine nucleotides. Dell'Aglio and Mhamdi [85] demonstrated that DHAR activity is extremely important for AsA recycling under low glutathione and high ascorbate concentrations, which were measured in sycamore germinating seeds (Figs 5 and 6). Increasing AsA/DHA ratio in germinating Norway maple embryonic axes (Fig 5) might indicate elevated AsA synthesis as well as effective enzymatic AsA regeneration (Fig 5). DHAR uses NAD(P)H and GSH as electron donors for DHA reduction. Both were available in comparable concentrations in *Acer* seeds, but the AsA/DHA ratio was still lower in sycamore seeds. DHAR activity was likely lower in sycamore seeds, as previously demonstrated during seed development and desiccation [21]. DHAR couples the Asc and glutathione pools to $H_2O_2$ [86]. In general, recalcitrant seeds have higher concentrations of ascorbate than orthodox seeds [86, 87]. The high accumulation of Asc in sycamore seeds is possibly a compensation strategy to survive injury related to germination-associated oxidative burst because a high Asc concentration protects against oxidative stress [88]. Another putative strategy to maintain redox balance in recalcitrant seeds might be keeping distinct $E_{GSSG/2GSH}$ in seed tissues (Fig 7), more precisely a highly reduced glutathione pool in embryonic axes and highly oxidized in cotyledons providing substrates to DHAR activity predominantly in embryonic axes.

Pyridine nucleotides together with ascorbate and glutathione enable redox homeostasis and cell cycle regulation [71]. Both Norway maple and sycamore seeds contain nuclei arrested at the 2C and 4C levels at maturity [57]. During stratification, nuclear DNA levels increase before radicle emergence uniquely in Norway maple seeds [57]. AsA is necessary for the G1 to S transition in the cell cycle [89, 90]. Increasing AsA/DHA ratio of Norway maple embryonic axes indicates that cell cycle arrest is progressively weakened in this species. Norway maple and sycamore seeds displayed distinct patterns of changes in Asc and glutathione redox couples during germination, reflecting their different physiology (Figs 5 and 6). Distinct strategies of redox regulation displayed in both species throughout germination eventually contributed to equilibrated $E_{GSSG/2GSH}$ at the beginning of the early development of sycamore seedlings (Fig 7). Desiccation tolerance is lost in orthodox seeds before the seedling stage [62, 63]. Thus, the redox control of plant growth associated with cell proliferation and cell differentiation is putatively more unified at the seedling stage, even originating from initially contrasting seeds. Interestingly, radicles protrusion at early seedling developmental stage is supported by *de novo* synthesis of GSH [28]. In this context, the growth of Norway maple radicles would be benefited by definitely higher GSH concentrations reported in embryonic axes at the germinated stage (Fig 6).

## Conclusions

Regulation of seed germination is complex and involves redox control based on consecutive reduction and oxidation reactions of molecules including Asc, glutathione and NAD(P). Norway maple and sycamore seeds are highly contrasted at the beginning of the germination process in terms of desiccation tolerance. This initial difference did not affect germination capacity,

which was high in both species. Sycamore germinated earlier than Norway maple, as dormancy in this species is not as deep as in Norway maple, as clearly confirmed by the changes in NAD. Similar to development and desiccation stages, at germination, Norway maple seeds displayed constant and efficient control of redox status in contrast to sycamore seeds, which displayed a distinct regulation of redox strategy status during germination (Fig 8). Glutathione, although lower in content in sycamore than in Norway maple seeds, was mostly reduced in sycamore seeds. Considerably higher Asc levels in sycamore counteracted oxidation processes enabling passage through all germination stages. Sycamore seeds exhibited a higher $NADPH/NADP^+$ ratio and thus higher reduction power, which drove anabolic and catabolic reactions dependent on NAD(P)H. Norway maple seeds, as benefited by multiple mechanisms enabling desiccation tolerance, operated germination longer, but in a more stable manner, producing high levels of NADPH, possibly via effective OPPP. In contrast, sycamore seeds germinated faster with more dynamic changes in redox status and efficient redox switches of pyridine nucleotides. The network of cellular redox agents is complicated and should concern the activity of proteins undergoing redox cycling, such as thioredoxins, glutaredoxins and peroxiredoxins, to obtain a full picture of redox state regulation during seed germination.

## Supporting information

**S1 Fig. Correlation matrices.** Correlation matrices calculated for embryonic axes (A) and cotyledons (B) of Norway maple, and embryonic axes (C) and cotyledons (D) of sycamore between concentrations of nicotinamide dinucleotide (NAD) phosphate (NADP) redox couples and their ratios, levels of ascorbate (Asc), ascorbic acid (AsA), dehydroascorbate (DHA), reduced (GSH) and oxidized (GSSG) glutathione, catabolic redox charge (CRC), anabolic redox charge (ARC), dormancy depth, phosphorylation capacity of isoform 1 (NADK1) and isoform 3 (NADK3) of NAD kinase, reducing power and half-cell reduction potential of glutathione ($E_{GSSG/2GSH}$) and ascorbate ($E_{DHA/AsA}$). Proportional data were transformed prior to analysis using the arcsine transformation. Crossed numbers indicate non-significant correlation (P > 0.05). (DOCX)

## Author Contributions

**Conceptualization:** Ewa Marzena Kalemba.

**Formal analysis:** Shirin Alipour, Karolina Bilska, Ewelina Stolarska, Natalia Wojciechowska.

**Investigation:** Shirin Alipour, Karolina Bilska, Ewelina Stolarska.

**Project administration:** Ewa Marzena Kalemba.

**Supervision:** Ewa Marzena Kalemba.

**Visualization:** Karolina Bilska, Natalia Wojciechowska, Ewa Marzena Kalemba.

**Writing – original draft:** Shirin Alipour, Ewa Marzena Kalemba.

**Writing – review & editing:** Shirin Alipour, Karolina Bilska, Natalia Wojciechowska, Ewa Marzena Kalemba.

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
