## [Decision Letter · Decision Letter 0]

4 Dec 2020

PONE-D-20-34053

Nicotinamide adenine dinucleotides are involved in distinct redox control of germination in Acer seeds with contrasting physiology

PLOS ONE

Dear Dr. Kalemba,

Thank you for submitting your manuscript to PLOS ONE. After careful consideration, we feel that it has merit but does not fully meet PLOS ONE’s publication criteria as it currently stands. Therefore, we invite you to submit a revised version of the manuscript that addresses the points raised during the review process.

I apologise for the slight delay in handling your manuscript. One of the reviewers was unable to provide me with their report so I reviewed your paper myself. 

We look forward to receiving your revised manuscript.

Kind regards,

Thomas Roach

Academic Editor

PLOS ONE

Additional Editor Comments:

Requested changes by the academic editor:

'involved' in the title is too strong because their involvement has not been shown. The data presented can be described as 'associated' or similar.

line 24. Please change the opening sentence. Germination is not a trait and the regulation of germination involves many other processes e.g. phosphorylation, metabolome. The role of transcription has also been questioned in studies where germination completed in the presence of transcription inhibitors.

line 29. '...between stratifying and germinating Acer seeds.'

line 38. plural not singular

line 59. 'consists of' rather than 'involves'.

line 70. how does it differ?

line 93. This is not yet sufficiently verified to be said as fact. 'The NAD/NADP ratio correlates with dormancy in Arabidopsis' or similar.

line 95. contrast what in seeds?

line 114. Putative means 'commonly accepted' which is not what is meant in this sentence.

line 129. This is a result and should be moved to line 193. It would also make it clearer in the results what is meant by 'germinated stage'.

line 142. No mention is made of what exactly is measured, e.g. a wavelength?

line 145. '...by its absorption at 265 nm in a...'

line 172. See line 93. Depth of dormancy would have to be verified with germination data at each stratification interval.

line 189. '...imbibition for 24 h halved the...'

line 234. a reference is missing after 'was reported'.

line 252. Unlcear. Sentence needs attention

line 255. Why radicles? There are no mention of them in materials and methods

line 265. see line 93.

line 266. Interpretation and needs moving to the discussion.

line 310. and 314. 'distinct' and 'dynamic' are not descriptive at all. Please improve

line 355. it is not clear what 'similarly' refers to here

line 358 and elsewhere. The variable of what is being compared is missing

line 377. 'reduction' here is presumably not redox-related and another word could be used to avoid confusion.

line 391 and 414. see line 114.

line 410. 'advanced' and not 'enhanced'

line 410. 'via' is too strong and can not be asserted

line 399-401. This sentence is not aligned to NADP data shown in Figure 2 and is confusing.

line 434-436. This sentence is not aligned to GSH data of the 2 species shown in Figure 5 and is confusing.

line 448. 'measured' rather than 'reported'

line 449. this statement not be supported because it excludes synthesis and breakdown

line 455-457. It needs mentioning that recalcitrant seeds have higher concentrations of ascorbate than orthodox seeds

line 466 This is not aligned to DHA data shown in Figure 4 and is confusing.

line 471. Individual tissues have not been measured

line 487. Too strong. It was not shown that these changes in the redox couples were required to accomplish germination

line 490. see line 114.

Line 491. 'saturation of electrons' is not appropriate

line 497. 'more' than what?

There are very few related publications on changes redox couples during seed germination, and authors are missing a few key references: i.e. on ascorbate (de Simone et al., 2017, https://core.ac.uk/download/pdf/83940636.pdf) and on glutathione (Gerna et al., 2017 https://pubmed.ncbi.nlm.nih.gov/28580817/)

Suggested additions / changes:

Did you make water content measurements of the seeds at each stage? It would be a very nice addition if you could calculate the half-cell reduction potentials for the redox couples (Schaffer and Buettner, 2001).

line 310. 'DO' as a abbreviation of 'degree of oxidation' is not often seen in the literature and is probably better not used

Journal Requirements:

"This research was supported by the Institute of Dendrology of the Polish Academy of Sciences."

Reviewers' comments:

Reviewer's Responses to Questions

**Comments to the Author**

1. Is the manuscript technically sound, and do the data support the conclusions?

Reviewer #1: Yes

2. Has the statistical analysis been performed appropriately and rigorously? 

Reviewer #1: Yes

3. Have the authors made all data underlying the findings in their manuscript fully available?

Reviewer #1: Yes

4. Is the manuscript presented in an intelligible fashion and written in standard English?

Reviewer #1: Yes

5. Review Comments to the Author

Reviewer #1: The comparison of the redox control of germination by the component of the ascorbate-glutathione cycle in desiccation tolerant and sensitive Acer species contributes to the better understanding of the regulation of this process. The presented results show that NAD(P) has an important role in the control of redox status during germination. The mode of this regulation is different in the two species. The manuscript is in general well written. The first part of the results [NAD(P)] and the discussion could be more concise. The similarities and differences in the redox control of germination between the two species could be summarized in a figure.

Remarks

1. l. 65: Regulating oxidative stress should be replaced by cellular redox environment.

2. l. 102: …in the seeds of the two Acer species…

3. l. 123: Why were seeds dried (D) to 10% water content (WC) for Norway maple and 30% WC for sycamore?

4. Figs. 1, 2, 4, 5: D, I and G should be explained in the legends.

5. 5. l. 191: …cotyledons of sycamore.

6. l. 213: The results for NADPH/NADP+ ratio should be shown later, after NADPH and NADP contents.

7. l. 241: The increase from 2.6 to 3 is not an exceptional one.

8. l. 276: The following title could be used: NAD(P)-originated physiological indices.

9. l. 298: …ratio except for D and I in embryonic axis…

10. l. 307: 4-fold instead of 40-fold.

11. l. 373 and 374: Use either NAD or NAD+!

12. l. 389: „and higher” is written twice.

6. PLOS authors have the option to publish the peer review history of their article (what does this mean?). If published, this will include your full peer review and any attached files.

Reviewer #1: No

---

## [Author Response · Author response to Decision Letter 0]

15 Dec 2020

Response to the Editor:

1. Requested changes by the academic editor:

'involved' in the title is too strong because their involvement has not been shown. The data presented can be described as 'associated' or similar.

Answer: Indeed, “involved is was too strong”. The title was changed as indicated.

line 24. Please change the opening sentence. Germination is not a trait and the regulation of germination involves many other processes e.g. phosphorylation, metabolome. The role of transcription has also been questioned in studies where germination completed in the presence of transcription inhibitors.

Answer: The opening sentence was changed as suggested. The second sentence was rewritten. Now it includes other possible regulation levels.

line 29. '...between stratifying and germinating Acer seeds.'

Answer: According to the definition of germination, germination sensu stricto begins with seed imbibition and ends with radicle protrusion. Stratification is the method of dormancy release, thus it is better to keep “germinating seeds”. Therefore we changed the names of X axes in all graphs to indicate that “2, 4, 6, 8, 10” are weeks after imbibition (WAI) in sycamore seeds and “3, 6, 9, 12” are WAI in Norway maple seeds. As we looked throughout literature related to seed germination mostly WAI or HAI (hours after imbibition) are used to indicate the time needed to complete germination process in seeds (lines 212-460 in revised manuscript). 

line 38. plural not singular

Answer: We changed it to plural (line 40 in revised manuscript).

line 59. 'consists of' rather than 'involves'.

Answer: 'involves' was replaced with 'consists of', as suggested (line 64 in revised manuscript). 

line 70. how does it differ?

Answer: Higher activity of enzymes involved in the ascorbate-glutathione cycle is a feature of orthodox Norway maple seeds. We modified the two sentences and changed them into:

“More precisely, orthodox seeds display higher activity of all enzymes involved in ascorbate-glutathione cycle. Redox homeostasis is essential for surviving desiccation and further germination [19,24–26].” (lines 75-77 in revised manuscript).

 line 93. This is not yet sufficiently verified to be said as fact. 'The NAD/NADP ratio correlates with dormancy in Arabidopsis' or similar.

Answer: We changed this sentence as suggested (line 102 in revised manuscript).

 line 95. contrast what in seeds?

Answer: The sentence was modified: “to contrast the state of the energy balance in orthodox and recalcitrant seeds” (line 104 in revised manuscript).

line 114. Putative means 'commonly accepted' which is not what is meant in this sentence.

R: Indeed, “putative” was replaced by “presumably” (line 124 in revised manuscript).

line 129. This is a result and should be moved to line 193. It would also make it clearer in the results what is meant by 'germinated stage'.

Answer: We added one sentence to clarify the G stage: “Seeds were assayed as germinated (G) when the radicle protruded to 5 mm above the seed testa.” The statement, assumed as results was given a reference to our results published this week, in which the dynamics of germination was investigated (lines 136-137 in revised manuscript).

line 142. No mention is made of what exactly is measured, e.g. a wavelength?

Answer: What exactly was measured is specified in the following subchapters 1-3 in which the wavelength is given (kinetic measurements at 600nm for NAD(P), absorbance at 265nm for ascorbate, kinetic measurements at 412 nm for glutathione) (lines 157-158 in revised manuscript).

line 156. '...by its absorption at 265 nm in a...'

Answer: This part was changed as suggested (line 165 in revised manuscript).

line 172. See line 93. Depth of dormancy would have to be verified with germination data at each stratification interval.

Answer: Depth dormancy expressed as the NAD/NADP ratio was quantified at each stratification interval and is given if Figure 4C.

line 189. '...imbibition for 24 h halved the...'

Answer: We added “for 24 h” to this sentence (line 208 in revised manuscript).

line 234. a reference is missing after 'was reported'.

Answer: This sentence was describing our results in Figure 2. However this sentence was deleted as description of NAD(P) results was suggested to be more concise (line 237 in revised manuscript).

line 252. Unlcear. Sentence needs attention

Answer: The sentence was shortened to focus on the main finding an not on a transient increase or decrease (line 272 in revised manuscript).

line 255. Why radicles? There are no mention of them in materials and methods

Answer: To be consistent with the nomenclature, “radicle” was replaced with “embryonic axis” (line 274 in revised manuscript).

line 265. see line 93.

Answer: We changed the description, “is” was replaced by “can be used as “ to indicate that it is possible and it is not as a fact (line 283 in revised manuscript).

line 266. Interpretation and needs moving to the discussion.

Answer: Indeed, We changed the place of these sentences and were moved to discussion section – last paragraph of Involvement of NAD(P) in dormancy regulation part (lines 430-431 in revised manuscript).

line 310. and 314. 'distinct' and 'dynamic' are not descriptive at all. Please improve

Answer: DO results were deleted from this manuscript and replaced by half-cell reduction potential, as suggested.

line 355. it is not clear what 'similarly' refers to here

Answer: 'similarly' referred to higher NAD(P) pool in orthodox seeds. However this word is unnecessary and was deleted.

line 358 and elsewhere. The variable of what is being compared is missing

Answer: The sentence was rewritten (lines 401-402 in revised manuscript).

line 377. 'reduction' here is presumably not redox-related and another word could be used to avoid confusion.

Answer: Indeed, “reduction” can be interpreted as changes in redox forms and not in concentration. We replaced “reduction” with “depletion of NAD concentration”. (line 428 in revised manuscript).

line 391 and 414. see line 114. 

Answer: According to Editor comment, this word was removed. (line 422 in revised manuscript). 

line 410. 'advanced' and not 'enhanced'

Answer: We introduced this change (line 461 in revised manuscript).

line 410. 'via' is too strong and can not be asserted

Answer: We replaced “via” with “and coincided with” (line 461 in revised manuscript).

line 399-401. This sentence is not aligned to NADP data shown in Figure 2 and is confusing.

Answer: The phrase “NADKs provide NADP” can explain higher NADP concentrations in Norway as a result of conversion from NAD. (NAD kinases phosphorylate NAD to NADP). This sentence was rewritten to indicate that in Acer seeds different NAD concentrations resulted into different concentrations of NADP, which is a substrate for glycolysis and PPP (lines 449-451 in revised manuscript).

line 434-436. This sentence is not aligned to GSH data of the 2 species shown in Figure 5 and is confusing.

Answer: This sentence refers to GSH data in germinated Norway maple seed (only last stage of germination) in which GSH level is considerably higher in Norway maple than in sycamore. We corrected this sentence (lines 496-497 in revised manuscript).

line 448. 'measured' rather than 'reported'

Answer: According to the Editor suggestion we replaced “reported” with “measured” (line 510 in revised manuscript).

line 449. this statement not be supported because it excludes synthesis and breakdown

Answer: This statement was rewritten to include also the effect of AsA syntheisis, not only enzymatic AsA regeneration, on AsA/DHA ratio (lines 511-512 in revised manuscript).

line 455-457. It needs mentioning that recalcitrant seeds have higher concentrations of ascorbate than orthodox seeds

Answer: According to the Editor suggestion we added new statement with the proper references. (lines 517-518 in revised manuscript).

line 466 This is not aligned to DHA data shown in Figure 4 and is confusing.

Answer: This sentence was rewritten. It concerns now on the ratio, not anymore on DHA, because its concentration was indeed not decreasing in Norway maple embryonic axes (line 529 in revised manuscript).

 line 471. Individual tissues have not been measured

Answer: This sentence was deleted as a consequence of not presenting the DO data.

line 487. Too strong. It was not shown that these changes in the redox couples were required to accomplish germination

Answer: We changed “to accomplish germination” into “during germination” (line 553 in revised manuscript).

line 490. see line 114.

Answer: The Editor is right and this word was deleted (line 555 in revised manuscript).

Line 491. 'saturation of electrons' is not appropriate

Answer: According to Editor suggestion we changed “saturation of electrons” to “reduction power” (lines 556-557 in revised manuscript).

line 497. 'more' than what?

Answer: We intended to state that is more complex then ascorbate, glutathione and NAD(P) which were investigated in our study. To make the sentence more clear we deleted this word. (line 562 in revised manuscript).

There are very few related publications on changes redox couples during seed germination, and authors are missing a few key references: i.e. on ascorbate (de Simone et al., 2017, https://core.ac.uk/download/pdf/83940636.pdf) and on glutathione (Gerna et al., 2017 https://pubmed.ncbi.nlm.nih.gov/28580817/)

Answer: Both literature positions are now described and cited in the Introduction section and Discussion section.

2. Suggested additions / changes: 

Did you make water content measurements of the seeds at each stage? 

Answer: The water content was measured and published in Wojciechowska eta al. 2020. This reference was added to the manuscript.

It would be a very nice addition if you could calculate the half-cell reduction potentials for the redox couples (Schaffer and Buettner, 2001).line 310. 'DO' as a abbreviation of 'degree of oxidation' is not often seen in the literature and is probably better not used

Answer: The half-cell reduction potentials for the redox couples of glutathione and ascorbate were calculated and presented in Figure 7. Therefore, DO were deleted because changes in DO are identical to changes in glutathione redox potential. The reduction potential (E) for the NADP+/NADPH half-cell is independent of their concentrations and depend on the ratio of the reduced and oxidized forms (Schaffer and Buettner, 2001). Therefore NAD(P)H/NAD(P)+ ratios were presented in Figure 3.

3. Journal Requirements:

3.1. Please ensure that your manuscript meets PLOS ONE's style requirements, including those for file naming. The PLOS ONE style templates can be found at

 Answer: We checked the additional requirements and added the Author contribution section to the manuscript.

3.2. Thank you for stating the following in the Acknowledgments Section of your manuscript:

"This research was supported by the Institute of Dendrology of the Polish Academy of Sciences."

Answers:

Conflicts of Interest section

Please do the changes in the online submission form. Please move the phrase "The authors declare no conflict of interest" to the "Conflicts of Interest" section. In the manuscript file this statement was in the correct section.

Authors order in the submission form

Please also change in the online submission form the order of the authors. During submission the corresponding author was set at first place and I couldn’t change it. Please move “Kalemba EM” for the last position. In the manuscript file the order of authors was correct.

Funding Section was changed in the manuscript as suggested:

“This research was funded by the National Science Center (Poland), grant No. 2015/18/E/NZ9/00729. The Institute of Dendrology of the Polish Academy of Sciences provided support in the form of salary and scholarships for all authors who are affiliated to the Institute. The specific roles of these authors are articulated in the ‘author contributions’ section. The funders had no role in study design, data collection and analysis, decision to publish, or preparation of the manuscript.”

Acknowledgments Section was deleted in the manuscript because information related to the support of The Institute was moved to the Funding Section.

Answers to the 3.2 part referring to the following sections: Conflicts of Interest, Funding and Acknowledgments were also included in the Cover Letter as it was suggested.

3.3. Please include captions for your Supporting Information files at the end of your manuscript, and update any in-text citations to match accordingly. Please see our Supporting Information guidelines for more information: http://journals.plos.org/plosone/s/supporting-information.

 Answer: Revised version of manuscript contains one supplemental figure and the figure caption of Fig S1 was added to the manuscript.

Response to the Reviewer 1:

Reviewer #1: The comparison of the redox control of germination by the component of the ascorbate-glutathione cycle in desiccation tolerant and sensitive Acer species contributes to the better understanding of the regulation of this process. The presented results show that NAD(P) has an important role in the control of redox status during germination. The mode of this regulation is different in the two species. The manuscript is in general well written. The first part of the results [NAD(P)] and the discussion could be more concise. The similarities and differences in the redox control of germination between the two species could be summarized in a figure.

 Answer: Description of results related NAD(P) was shortened. A scheme contrasting two different strategies of redox regulation of germination in orthodox and recalcitrant seeds was created and added to the manuscript as Figure 8.

1. line 65: Regulating oxidative stress should be replaced by cellular redox environment.

Answer: According to Reviewer suggestion this part was corrected (line 70 in revised manuscript).

2. line 102: …in the seeds of the two Acer species…

Answer: The changes was done (line 112 in revised manuscript).

3. line 123: Why were seeds dried (D) to 10% water content (WC) for Norway maple and 30% WC for sycamore?

Answer: Dehydration of sycamore seeds, categorized as recalcitrant seeds, below a safe range of water content (27% WC) leads to water stress and significant decrease in viability because recalcitrant seed are sensitive to desiccation. Desiccation of Norway maple seeds, categorized as orthodox, to 8–10% causes no changes in their viability (Hong and Ellis 1990; Pukacka and Czubak 1998).

4. Figs. 1, 2, 4, 5: D, I and G should be explained in the legends.

Answer: Explanations of abbreviations used in graphs were added in the figure captions.

5. line 191: …cotyledons of sycamore.

Answer: We do not agree with this comment. In this part, we presented results of NAD level in Norway maple seeds and comparing them between embryonic axes and cotyledons.

6. line 213: The results for NADPH/NADP+ ratio should be shown later, after NADPH and NADP contents.

Answer: Thank you for this comment. Indeed, it should be NADH/NAD+ ratio. Therefore, we changed it to the correct form and not moved to other section. (line 252 in revised manuscript).

7. line 241: The increase from 2.6 to 3 is not an exceptional one.

Answer: “Exceptionally high” was replaced by “the highest” (line 253 in revised manuscript).

8. line 276: The following title could be used: NAD(P)-originated physiological indices.

Answer: According to Reviewer suggestion we changed to “NAD(P)-originated physiological indices” (line 295 in revised manuscript).

9. line 298: …ratio except for D and I in embryonic axis…

Answer: Since in this sentence we describe germination stages, we added “except imbibition”, because germination do not include dry stage, it begins with imbibition. (line 318 in revised manuscript).

10. line 307: 4-fold instead of 40-fold.

Answer: We corrected it. (line 326 in revised manuscript).

11. line 373 and 374: Use either NAD or NAD+!

Answer: “NAD” is used when the describe the whole NAD pool consisting of reduced (NADH) and oxidized (NAD+) forms. Therefore NAD+ referes to measured concentrations of the oxidized form only.

12. line 389: „and higher” is written twice.

Answer: The Reviewer is right. We deleted one of them. (line 440 in revised manuscript).

---

## [Decision Letter · Decision Letter 1]

5 Jan 2021

Nicotinamide adenine dinucleotides are associated with distinct redox control of germination in Acer seeds with contrasting physiology

PONE-D-20-34053R1

Dear Dr. Kalemba,

We’re pleased to inform you that your manuscript has been judged scientifically suitable for publication and will be formally accepted for publication once it meets all outstanding technical requirements.

Kind regards,

Thomas Roach

Academic Editor

PLOS ONE

Additional Editor Comments (optional):

Reviewers' comments:

Reviewer's Responses to Questions

**Comments to the Author**

1. If the authors have adequately addressed your comments raised in a previous round of review and you feel that this manuscript is now acceptable for publication, you may indicate that here to bypass the “Comments to the Author” section, enter your conflict of interest statement in the “Confidential to Editor” section, and submit your "Accept" recommendation.

Reviewer #1: All comments have been addressed

2. Is the manuscript technically sound, and do the data support the conclusions?

Reviewer #1: Yes

3. Has the statistical analysis been performed appropriately and rigorously? 

Reviewer #1: Yes

4. Have the authors made all data underlying the findings in their manuscript fully available?

Reviewer #1: Yes

5. Is the manuscript presented in an intelligible fashion and written in standard English?

Reviewer #1: Yes

6. Review Comments to the Author

Reviewer #1: The authors made the recommended modifications during the revision of their manuscript or gave an appropriate answer for my remarks.

7. PLOS authors have the option to publish the peer review history of their article (what does this mean?). If published, this will include your full peer review and any attached files.

Reviewer #1: No

---

## [Editor Report · Acceptance letter]

18 Jan 2021

PONE-D-20-34053R1 

Nicotinamide adenine dinucleotides are associated with distinct redox control of germination in *Acer* seeds with contrasting physiology 

Dear Dr. Kalemba:

I'm pleased to inform you that your manuscript has been deemed suitable for publication in PLOS ONE. Congratulations! Your manuscript is now with our production department. 

Kind regards, 

on behalf of

Dr. Thomas Roach 

Academic Editor

PLOS ONE